# Beclin1 Binds to Enterovirus 71 3D Protein to Promote the Virus Replication

**DOI:** 10.3390/v12070756

**Published:** 2020-07-14

**Authors:** Qi Xiang, Pin Wan, Ge Yang, Siyu Huang, Mengying Qin, Hua Yang, Zhen Luo, Kailang Wu, Jianguo Wu

**Affiliations:** 1State Key Laboratory of Virology, College of Life Sciences, Wuhan University, Wuhan 430072, China; 2017202040043@whu.edu.cn (Q.X.); 2018102040008@whu.edu.cn (G.Y.); 2015202040054@whu.edu.cn (S.H.); 2017202040030@whu.edu.cn (M.Q.); 2014102040012@whu.edu.cn (H.Y.); 2Guangdong Provincial Key Laboratory of Virology, Institute of Medical Microbiology, Jinan University, Guangzhou 510632, China; 2012202040066@whu.edu.cn (P.W.); zhluo18@jnu.edu.cn (Z.L.)

**Keywords:** enterovirus 71, EV71, EV71 non-structural protein 3D, Beclin1, autophagy

## Abstract

Enterovirus 71 (EV71) is the main pathogen causing hand-foot-mouth disease (HFMD) in infants and children, which can also lead to severe neurological diseases and even death. Therefore, understanding the replication mechanism of EV71 is of great significance for the prevention and control of EV71-induced diseases. Beclin1 (BECN1, a mammalian homologue of ATG6 in yeast) is an important core protein for the initiation and the normal process of autophagy in cells. In addition to its involvement in autophagy, Beclin1 has also been reported to play an important role in cancer and innate immune signaling pathways. However, the role of Beclin1 in EV71 replication remains elusive. Here, we primarily found that Beclin1 facilitates EV71 replication in human rhabdomyosarcoma (RD) cells and the autophagy was actually induced, but Beclin1 was not significantly affected at either mRNA level or protein level during early EV71 infection. Further studies discovered that Beclin1 could interacts with EV71 non-structural protein 3D mainly through its evolutionary conserved domain (ECD) and coiled-coiled domain (CCD), thus promoting the replication of EV71 in human rhabdomyosarcoma (RD) cells and human astroglioma (U251) cells. Collectively, we reveal a novel regulatory mechanism associated with Beclin1 to promote EV71 replication, thus providing a potential therapeutic target for the prevention and control of EV71-associated diseases.

## 1. Introduction

Enterovirus 71 (EV71) is a single-stranded positive RNA virus of the Picornaviridae family with a genome of approximately 7.5 kb. The EV71 genome encodes a long polyprotein with a single open reading frame followed by a polyA tract. The single polyprotein is flanked by untranslated regions at both the 5′ and 3′ ends and can be divided into three different genomic regions (P1, P2 and P3). The P1 genomic region encodes the structural proteins VP1–VP4. The P2 and P3 genomic regions encode the nonstructural proteins 2A, 2B, 2C, 3A, 3B, 3C and 3D [1]. EV71 causes hand, foot and mouth disease (HFMD) and can even cause fatal central nervous system (CNS) infections in young children, including encephalitis, aseptic meningitis and acute flaccid paralysis [2]. Since it was first identified from a child with neurologic symptoms in California in 1969 [3], epidemic and sporadic outbreaks of neurovirulent EV71 infections have been reported worldwide especially in the Asia-Pacific region [4,5,6,7,8,9]. It is highlighted that inactivated EV71 vaccine elicited EV71-specific immune responses and protection against EV71-associated HFMD [10]. However, the pathogenicity of EV71 is not entirely understood, and there are no effective antiviral treatments to control and prevent its outbreak.

The innate immune system, serving as the first line of defense against foreign pathogen infection, plays a significant role in controlling viruses. However, EV71 has evolved and developed different strategies to evade the innate immune system in order to facilitate replication inside the host cell. Previous studies have proved that EV71 nonstructural proteins 2A, 2C, and 3C could inhibit interferon response by cleaving several key proteins in this pathway, such as TRIF, IRF7, MAVS and IFNR1 [11,12,13,14,15,16,17]. Furthermore, EV71 infection induces the upregulation of USP19 which negatively regulates cellular antiviral type I interferon signaling by targeting the tumor necrosis receptor-associated factor 3 (TRAF3) molecule and decreasing TRAF3 ubiquitination of K63-linkage [18]. Enterovirus-induced miR-146a also facilitates viral pathogenesis by suppressing IFN production [19,20]. Hence, discovery of new mechanism in EV71 replication is of great significance for control of EV71 infection.

Beclin1, the mammalian orthologue of yeast Atg6, is one of the earliest characterized mammalian autophagy proteins [21], initially identified as a Bcl2-binding protein [22]. Beclin1 plays a pivotal role in autophagy initiation [23], which governs the autophagy process by regulating PI3K complex maturation and the subsequent recruitment of additional Atg proteins to orchestrate autophagosome formation [24,25,26,27]. BECN1 is also an essential gene required for embryonic survival and development [28], but also for the development of cancer and several neurodegenerative diseases [21,29,30,31]. Some viruses target Beclin1 to evade autophagy degradation, and these include the human immunodeficiency virus (HIV) [32], influenza A virus [33], African swine fever virus [34], foot and mouth disease virus [35], a-herpesvirus (aHV) herpes simplex virus type 1 (HSV-1) [36], and human cytomegalovirus (HCMV) [37]. However, the relationship of EV71 and Beclin1 remains elusive.

In this study, we firstly revealed that autophagy was actually induced, but Beclin1 was not significantly affected at either mRNA level or protein level in rhabdomyosarcoma (RD) cells infected with EV71. Next, we found that overexpression of Beclin1 could promote EV71 replication in both a dose-dependent and time-dependent manner. Subsequently, we constructed RD cells and U251 cells that stably interfered with Beclin1 and discovered that EV71 replication was significantly inhibited. Further studies revealed that Beclin1 could interact with EV71 non-structural protein 3D, and the coiled-coiled domain (CCD) and evolutionary conserved domain (ECD) of Beclin1 are indispensable for promoting the replication of EV71. In conclusion, we found a new function of Beclin1 in viral infection and revealed a new regulatory mechanism to enhance EV71 replication, thus providing a potential therapeutic target for the prevention and control of EV71-associated diseases. 

## 2. Materials and Methods

### 2.1. Cell Lines, Viruses, Transfection and Infection

Human embryonic kidney (HEK293T) cells, human rhabdomyosarcoma (RD) cells, and human astroglioma (U251) cells were purchased from the China Center for Type Culture Collection (CCTCC, Wuhan, China). Cells were cultured in Delbecco modified Eagle medium (DMEM) (Gibco, Grand Island, NY, USA) supplemented with 10% FBS, 100 U/mL penicillin, and 100 μg/mL streptomycin at 37 °C under 5% CO_2_. Cells were transfected with PEI or Lipofectamine 2000 (Invitrogen, Carlsbad, CA, USA) according to the manufacturer’s instructions. Enterovirus 71 Xiangyang strain (GenBank accession number JN230523.1) isolated by our group was used in this study. Virus infection was carried out as described previously [38]. Briefly, cells were infected with EV71 at the indicated multiplicities of infection (MOI) in serum-free medium and washed twice in PBS after two hours incubation. Cells were then maintained with fresh medium supplemented with 2% FBS for the following experiments. 

### 2.2. Antibodies and Reagents

Mouse anti-Flag (F1804) and rabbit anti-HA (H6908) antibodies were purchased from Sigma-Aldrich (St Louis, MO, USA). Rabbit anti-GFP (50430-2-AP), Rabbit anti-LC3 (14600-1-AP) and Rabbit anti-P62 Polyclonal antibody(18420-1-AP) were purchased from Proteintech Group (Chicago, IL, USA). Rabbit anti-β-actin (AC026) and anti-EV71-3D polyclonal antibodies were produced by ABclonal Technology (Wuhan, China). Rabbit anti-EV71-VP1 (PAB7631-D01P) polyclonal antibody was purchased from Abnova (Taipei city, Taiwan). Rabbit anti-Beclin1 (381896) antibody was purchased from ZENBIO (Chengdu, China). Delbecco modified Eagle medium (DMEM), and fetal bovine serum (FBS) were purchased from Gibco (Grand Island, NY, USA). Lipofectamine 2000, normal rabbit IgG and normal mouse IgG were purchased from Invitrogen Corporation (Carlsbad, CA, USA). Protein ladder (26616) was purchased from Thermo scientific (Rockford, IL, USA). Complete, EDTA-free Protease Inhibitor Cocktail was purchased from Roche (Basel, Switzerland). Chloroquine diphosphate were purchased from Selleck (Houston, TX, USA). Rapamycin (53123-88-9) was obtained from Target Mol (Shanghai, China).

### 2.3. Plasmids Construction and Primers

The cDNA regions of human Beclin1 were generously provided by the Han Jiahuai laboratory of Xiamen University and then cloned into pcDNA3.1 (+)-3× Flag and pCAGGS-HA vectors. The Beclin1 mutants Flag-Beclin1-BD, Flag-Beclin1-BD+CCD, Flag-Beclin1-BD+CCD+ECD, Flag-Beclin1-CCD+ECD, and Flag-Beclin1-ECD, were all constructed into pcDNA3.1 (+)-3× Flag vector. To construct plasmids expressing EV71 nonstructural protein 3D, the coding sequence of 3D was cloned into pCAGGS-HA and pEGFP-C1 vectors.

### 2.4. Quantitative RT-PCR Analysis and Primers

Total RNA was extracted from cells using TRIzol reagent (Invitrogen Life Technologies; Carlsbad, CA, USA) according to the manufacturer’s instructions. RNA was reversed by HiScript II Q RT Supermix (Vazyme Biotech Co, Nanjing, China) to obtain complementary DNA and the following quantitative real-time analysis was performed using ChamQ SYBR qPCR Master Mix (Vazyme Biotech Co, Nanjing, China) on a Roche LC480 (Roche Diagnostics; Penzberg, Germany). All testing primers were designed by Primer Premier 5.0 and their sequences were as follows: EV71-VP1 forward: 5′-AATTGAGTTCCATAGGTG-3′, and EV71-VP1 reverse: 5′-CTGTGCGAATTAAGGACAG-3′; BECN1 forward: 5′-CCATGCAGGTGAGCTTCGT-3′, and BECN1 reverse: 5′-GAATCTGCGAGAGACACCATC-3′, GAPDH forward: 5′-GGAGCGAGATCCCTCCAAAAT-3′, and GAPDH reverse: 5′-GGCTGTTGTCATACTTCTCATGG-3′.

### 2.5. 50% Tissue Culture Infective Dose (TCID50) Assay

RD cells were grown in 96-well plates to 90% confluence and inoculated with the supernatants from infected RD cells for 2 h. Cells were washed with phosphate buffer saline (PBS) and cultured with 2% FBS in DMEM. At day three post-infection, the plates were examined for the lowest dilution at which 50% of the wells showed the cytopathic effect using the Reed-Muench method.

### 2.6. Lentivirus Production and Generation of Stable Cell Lines

The targeting sequences of short hairpin RNA (shRNA) for the human BECN1 were: shBECN1-1: 5′-CCCGTGGAATGGAATGAGATT-3′, shBECN1-2: 5′-CTCAAGTTCATGCTGACGAAT-3′. After annealing with the complementary shRNA, the double stranded shRNA for BECN1 was ligated into the pLKO.1 vector (Sigma-Aldrich; St. Louis, MO, USA). A pLKO.1 vector encoding shRNA for a negative control or BECN1 was transfected into HEK293T cells together with psPAX2 (#12260, Addgene, Watertown, MA, USA) and pMD2.G (#12259, Addgene, Watertown, MA, USA) by Lipofectamine 2000 (Invitrogen, Carlsbad, CA, USA). Culture supernatants were harvested at 36 h and 48 h after transfection and then centrifuged at 2000 rpm for 10 min plus, filtrating the supernatants through a 0.45 μm filter to remove the cells. RD and U251 cells were infected with the collected supernatants containing lentiviral particles in the presence of 8 μg/mL polybrene (Sigma, Saint Louis, MO, USA). After 48 h of cultivation, cells were selected by 2.5 μg/mL puromycin (Sigma, Saint Louis, MO, USA) for seven days continuously. Knockdown efficiency of shRNA-targeted BECN1 was identified by RT-PCR and immunoblot analyses.

### 2.7. Western Blot Analysis and Co-Immunoprecipitation Assays

For Western blot, cells were washed twice in ice-cold PBS and then lysed in lysis buffer (20 mM Hepes, pH7.4, 150 mM NaCl, 1 mM EDTA, 1 mM EGTA and 1% Triton-X100) in the presence of cocktail (Roche) on ice. Protein concentration was determined by Bradford assay (Bio-Rad, Hercules, CA, USA). After adding 5× SDS loading buffer, cell lysates were electrophoresed by 10–12% SDS-PAGE gel and then transferred onto a nitrocellulose membrane (Millipore Sigma-Aldrich, St. Louis, MO, USA). The membranes were blocked in 5% nonfat dried milk dissolved in PBS with 0.1% Tween-20 and incubated with specific primary antibodies overnight at 4 °C. After incubating with the secondary antibodies at room temperature for 2 h, protein bands were detected with the Clarity Western ECL substrate (Bio-Rad) and Luminescent Image Analyzer (LAS-4000, Fujifilm, Tokyo, Japan). 

For Co-IP assays, cells were washed with cold PBS and lysed in 1 mL RIPA buffer (50 mM Tris-HCl, pH7.4, 300 mM NaCl, 1% NP-40, 1 mM EDTA, and 5% glycerol) containing protease inhibitors. 100 μL of the lysates were subjected for immunoblot analysis to detect the expression of target proteins. The rest of the lysates were incubated with a control IgG or the indicated primary antibodies at 4 °C overnight and were further incubated with protein A/G-agarose (GE Healthcare, Waukesha, WI, USA) for 3–4 h. The beads were washed five times by washing buffer (50 mM Tris-HCl, pH7.4, 500 mM NaCl, 1% NP-40, 1 mM EDTA, and 5% glycerol) and resuspended in 50 mL 2× SDS loading buffer before Western blot analysis.

### 2.8. Immunofluorescence Analysis

Processed cells were washed with PBS containing 0.1% BSA and fixed with 4% paraformaldehyde for 15 min at room temperature, followed by permeabilization with 0.2% Triton-X100 for 5 min, and blocked with 5% BSA for 1 h. Anti- Flag and HA or anti-3D antibodies were added and incubated at 4 °C subsequently, and then FITC-conjugated anti-mouse and Cy3-conjugated anti-rabbit secondary antibodies were incubated with cells for 45 min away from light. Eventually, cells were washed three times with PBS containing 0.1% BSA and stained with DAPI for 5 min at 37 °C, and confocal imaging was performed using Fluo View FV1000 (Olympus, Tokyo, Japan).

### 2.9. Statistical Analyses

All experiments were repeated at least three times with similar results. All data were expressed as the mean ± SD (standard deviation). Statistical analysis was carried out using the Student’s *t* test for two groups and one-way ANOVA for multiple groups (GraphPad Prism 6.0, Inc. La Jolla, CA, USA). Differences were considered statistically significant when *p* < 0.05. 

## 3. Results

### 3.1. The Expression Level of Beclin1 Was Stably Maintained during EV71 Infection

When studying Beclin1 as a core protein for the initiation and normal process of autophagy in cells, we firstly verified whether autophagy was induced during EV71 infection. A dual fluorescent tagged-plasmid GFP-mCherry-LC3 was used to indicate the course of autophagic flux by confocal assay as previously reported [39]. It is well known that LC3-attached autophagosomes are formed in the cytoplasm when autophagy is induced and then fuse with endosomes or lysosomes to form autolysosomes, which provide an acidic environment and digestive function to the interior of the autophagosome. As the mCherry is acid stable while GFP is acid labile, so, if autophagic flux is increased, both yellow and red punctate are increased; however, if autophagosome maturation into autolysosomes is blocked, only yellow punctate is increased without a concomitant increase in red punctate. We found EV71 infection induced autophagy at 8 and 12 h post-infection in RD cells (Figure 1A,B). As a positive control, it is apparently that autophagy was activated by Rapamycin (RAP) and autolysosomes were blocked by chloroquine (CQ) (Figure 1A,B), which is similar as previously reported [40,41]. To further confirm the influence of EV71 on Beclin1 during infection, we infected RD cells with EV71 in a time-dependent manner or dose-dependent manner. In EV71-infected RD cells, the levels of EV71 VP1 RNA (Figure 1D,H) and 3D protein (Figure 1E,I) were found to be significantly increased at each indicated time point or MOI post-infection as expected. However, the *BECN1* RNA level (Figure 1C,G) and Beclin1 protein level (Figure 1E,I) were not significantly altered. Previous studies have illustrated that EV71 infection led to the upregulation of USP19 [18] and USP19 could stabilize Beclin1 by decreasing the K11-linked ubiquitin chains leading to the inhibition of Beclin1 degradation [42], which may provide partial explanation of our results. Besides, we measured the ratio of LC3-I to LC3-II and the protein levels of a well characterized autophagic substrate, p62/SQSTM1, which binds to LC3 and is specifically degraded as a result of complete autophagic flux; the conversion to LC3-II increased and the level of the p62 was reduced with the increase of infection time and infection concentration, indicating that autophagy was induced and promoted upon EV71 infection. In conclusion, these results suggested that EV71 infection induced autophagy and maintained the expression level of Beclin1.

### 3.2. Overexpression of Beclin1 Promotes EV71 Replication in RD Cells

To explore the role of Beclin1 in EV71 infection, RD cells were transfected with Flag-tagged Beclin1 by an increased degree; at 24 h post-transfection, cells were infected with EV71 at a MOI = 1 for another 12 h. The capsid protein of EV71-VP1, which is usually used to identify the replicating efficiency of EV71, was significantly enhanced in both mRNA level (Figure 2A) and protein level (Figure 2B), which indicated that Beclin1 was able to promote EV71 replication. Furthermore, we also demonstrated that EV71 replication was promoted by overexpression of Beclin1 in a time-dependent manner (Figure 2C,D). Collectively, the results of Beclin1 overexpression in EV71-infected RD cells proved that Beclin1 could promote EV71 replication in host cells.

### 3.3. Knockdown of BECN1 Attenuates EV71 Replication in RD and U251 Cells

Since the overexpression of Beclin1 can strengthen EV71 replication and EV71 infection does not change the expression level of Beclin1, we next identified the effects of knockdown of BECN1 on EV71 infection. Firstly, we synthesized two different BECN1(Beclin1)-targeted shRNAs and transfected into RD cells so as to identify the knockdown efficiency and its influence on EV71 replication. The results show that the mRNA level and protein level of Beclin1 were significantly reduced in RD cells; moreover, knockdown of BECN1 inhibited EV71 replication (Figure 3A,B). In addition, supernatants from infected RD cells were transfected with Flag-vector or Flag-Beclin1, pLKO.1-shNC or pLKO.1-shBeclin1 for 12 h and then collected for TCID_50_ assay. Viral TCID_50_ was obviously upregulated in Beclin1 overexpressed cells compared with mock cells, while viral titer was significantly reduced in Beclin1-knockdown cells compared to Beclin1 normally expressed cells, further indicating that Beclin1 promoted EV71 replication (Figure 3C). Next, we utilized lentivirus to establish stable BECN1-knockdown RD cells and U251 (human astroglioma) cells, then we infected stable RD and U251 cells with EV71 and checked replication efficiency at different time points in two different cells. We discovered that the mRNA level and protein level of EV71-VP1 were remarkably decreased compared with responding controls both in RD (Figure 3D,E) and U251 (Figure 3G,H) cells. Interestingly, on account of Beclin1 playing a pivotal role in autophagy [43,44,45], we have also tested LC3 (a marker of autophagy) change after knockdown of BECN1 (Figure 3F,I). It was shown that the LC3-II/LC3-I ratio was attenuated when Beclin1 interfered, which may indicate that autophagy flux was involved in Beclin1 enhancing EV71 replication. Some studies have reported the relationship between EV71 and autophagy [40,41,46]; however, the molecular mechanism is still indistinct. With all the above results, we firstly provided the evidence that Beclin1 promoted EV71 infection in RD and U251 cells.

### 3.4. EV71 Nonstructural Protein 3D Interacts and Co-Localizes with Beclin1 in the Cytoplasm

To further evaluate the relationship between EV71 and Beclin1, and the possible mechanism utilized by Beclin1, we screened the interactions among EV71 nonstructural proteins and Beclin1. Co-immunoprecipitation (Co-IP) assay was carried out as previously described [47]. Briefly, plasmid encoding Flag-Beclin1 and plasmids encoding each of the EV71 non-structure proteins, HA-2A, HA-2B, HA-2C, HA-3AB, HA-3C, HA-3D and Flag-Beclin1, were co-transfected into Human embryonic kidney (HEK293T) cells. Co-IP results showed that EV71 3D specifically interacted with Beclin1, whereas other EV71 viral proteins failed to interact with Beclin1 (Figure 4A). Additionally, we repeated the Co-IP assay by co-transfecting 3D and Beclin1 in HEK293T cells. Further results confirmed the interaction between 3D and Beclin1 (Figure 4B). Immunofluorescence analysis also showed that 3D and Beclin1 could co-localize in the cytoplasm (Figure 4C). Moreover, RD cells were infected with EV71 at a MOI = 2 for 12 h and cell lysates were immunoprecipitated with anti-Beclin1 antibody. It is shown that there was an endogenous interaction between EV71-3D and Beclin1 (Figure 4D). Altogether, these results consistently revealed the interaction between 3D and Beclin1.

### 3.5. The Beclin1 CCD and ECD Domains Were Essential for the Facilitation of EV71 Replication

To further evaluate the region of Beclin1 responsible for interacting with 3D and facilitating EV71 replication, a series of mutants were constructed based on the functional domains of Beclin1, including Beclin1 mutants Beclin1-BD, Beclin1-BD+CCD, Beclin1-BD+CCD+ECD, Beclin1-CCD+ECD, and Beclin1-ECD (Figure 5A). To determine that the domain of Beclin1 is responsible for interacting with 3D, plasmids expressing GFP-3D and Flag-Beclin1 or various mutants were transfected into HEK293T cells. Co-IP assay revealed that Beclin1 coprecipitated with 3D, whereas this interaction was weakened when CCD and ECD domains were deleted (Figure 5B), suggesting CCD and ECD domains were indispensable in the interaction between Beclin1 and 3D. To further assess the functional domain of Beclin1 for promoting EV71 replication, plasmids expressing Flag-vector, Flag-Beclin1 and various mutants were transfected into RD cells followed by EV71 infection. Subsequently, Real-time PCR and Western blot results showed that CCD and ECD domains were essential for Beclin1 to maintain its function in the promotion of EV71 replication (Figure 5C,D). Thus, Beclin1 CCD and ECD domains were essential for the interaction between Beclin1 and 3D in promoting EV71 replication.

## 4. Discussion

Enterovirus 71 (EV71) is the major pathogen of hand-foot-mouth disease (HFMD), which affects mainly children of age five and below [48,49]. Unlike coxsackieviruses, another member of Picornaviridae family, also leading to HFMD [50,51], EV71 infections are usually associated with severe neurological diseases such as poliomyelitis-like paralysis, aseptic meningitis and encephalitis [52], which accounts for 70% of severe HFMD cases and 90% of HFMD-related deaths [53]. Although many vaccines and antiviral drugs against EV71 have been developed [54,55,56], there are still challenges involving the worldwide epidemic of EV71, including their applicability against various EV71 pandemic strains in other countries, international requirements on vaccine production and quality control, etc. Therefore, it is highly important to discover new mechanisms underlying host regulating EV71 infection and find new potential therapeutic targets against EV71. Our group has previously reported several host factors regulating EV71 replication; for example, SIRT1 can inhibit EV71 replication and RNA translation by interfering with the viral polymerase and 5′UTR RNA [57], and PolyC-Binding Protein 1 interacts with 5′-untranslated region of EV71 RNA in membrane-associated complex to facilitate viral replication [58]. All of these findings provide new insights into the development of anti-EV71 therapeutic strategies.

Beclin1 plays a central role in the autophagy [23,59]. In addition, it has also been reported that Beclin1 plays an important role in viral replication. Coronavirus membrane-associated papain-like proteases induce incomplete autophagy through interacting with Beclin1 and negatively regulate antiviral innate immunity to promote viral replication [60]. The Nef protein of HIV functions in preventing destruction of HIV components in autolysosomes by interacting with Beclin1, thus shielding HIV from autophagy in its role of a cell autonomous antimicrobial defense [32]. In FMDV (Foot-and-mouth disease virus)-infected cells, the largest viral protein in the replication complex, 2C, would bind to Beclin1 to prevent the fusion of lysosomes to autophagosomes, allowing for virus survival [35]. Beclin-1 has also been shown to interact with cGAS or MAVS to suppress cGAMP synthesis or RIG-I-MAVS interaction, halting type I interferon (IFN) production upon recognition of external pathogens [42,61]. Nevertheless, the mechanisms of Beclin1 in EV71 replication is not clearly known and EV71-mediated effects on Beclin1 are not reported. 

In our study, we firstly identified a novel role of Beclin1 in EV71 infection and identified Beclin1 as an essential component of the autophagy pathway to EV71. When EV71 infects host cells, Beclin1 will bind with 3D protein and promote the replication of EV71. Furthermore, early EV71 infection has few influences on the protein and mRNA level of Beclin1. The different domains of Beclin1, shown in Figure 5A, mediate communications with multiple interaction partners that can alter its conformation and binding accessibility, thus making Beclin1 an important molecular platform for the regulation of PI3K activity and autophagy. Beclin1 interacts with the PI3K core component VPS34 and lipid membranes via its evolutionary conserved domain (ECD) in C-terminal [43,62] and binds to ATG14L or UVRAG in a mutually exclusive manner via its coiled-coiled domain (CCD) [63]. The BH3 domain (BD) of Beclin1 mediates its interaction with the antiapoptotic protein BCL2 and this interaction causes a steric block inhibiting PI3K complex formation [64], and AMBRA1 is a positive regulator of autophagy and competes with BCL2 for binding to the Beclin1 BH3 domain [65,66]. Interestingly, we found that the CCD and ECD domains of Beclin1 are most effective for interacting with 3D and promoting EV71 replication, but it is hard to clearly understand why these two domains play the most important role.

EV71 3D protein is an RNA-dependent RNA polymerase and is responsible for the genomic RNA replication of EV71 [67,68]. A previous study has clarified that EV71-activated SIRT1 binds with the 3D protein and attenuates the acetylation and RdRp activity of 3D, resulting in the repression of viral genome replication [57]. Beclin1 could assist viral reproduction through impacting the RdRp activity of 3D. Some studies have also reported other functions of 3D; one research study reported that EV71 mediated cell cycle arrest in S phase through 3D [69], and another claimed that 3D protein facilitates the release of IL-1β by binding with NLRP3 and enhancing the assembly of inflammasome complex [70]. Recently, a new finding pointed out that EV71 3D inhibits MDA5-mediated IFN-β promoter activation via interacting with the CARD domain of MDA5 protein [71]. Our study may provide further evidence for 3D inhibiting IFN responses and promoting viral replication. EV71-3D makes use of the interaction with Beclin1 to suppress type I IFN signaling pathway due to Beclin1 acting as a negative regulator of RIG-I-MAVS mediated IFN response [42]. However, the precise molecular mechanism is still indistinct. In addition, there may be effects other than the interaction of Beclin1 alone with the 3D protein involved, as shown in Figure 3. When Beclin1 interfered, we found that higher levels of LC3-II were accumulated in mock-cells than in Beclin1 knockdown-cells upon EV71 infection, and higher levels of EV71 were replicated in mock-cells than in Beclin1 knockdown-cells, which may indicate that higher EV71 replication is positively correlated to a higher level of autophagy, because knockdown of Beclin1 is likely to alter functions of other host proteins of the autophagy pathways and weaken the dynamism of the autophagy [72]. Even though EV71-induced autophagy and its pro-viral role have been documented [41], disagreement still exists [46] and which components of the autophagy pathway are used still need to be determined. In fact, our research might have discovered one of the specific elements involved in EV71 infection, i.e., EV71 possibly propels 3D to interact with Beclin1 in order to regulate the process of autophagy so as to achieve the purpose of massive viral proliferation. However, a finding published in 2019 demonstrated that the knockout of Beclin1 has no effect on replication of Poliovirus (PV), and PV belongs to the enteroviruses, like EV71 [73]. In our opinion, each virus, even closely related viruses like PV and EV71, have different structural features and life cycles, and cells used in experiments are also different. Furthermore, this paper has also illustrated that even DENV and ZIKV, both of which are members of flavivirus, will use an exclusive component of the autophagy pathway for their own replication. Therefore, it is not incompatible that each virus uses slightly different components. A human genome-wide RNAi screen carried out in 2016 has also identified that Beclin1 is involved in EV71 replication in human rhabdomyosarcoma cells [74], which is consistent with our results. 

As mentioned above, autophagy was induced to offer a favorable micro-environment for EV71’s propagation within an infected host, and treatment of cells with tamoxifen, rapamycin and serum starvation, which are agonists of autophagy, promoted viral yields, while inhibition of autophagy by 3-MA, bafilomycin A1 or saikosaponin D, suppressed viral production [40,41,75]. However, it is unknown whether EV71 induces autophagic flux, since no reports have actually shown that the number of autolysosomes is increased in EV71 infected cells. Interestingly, our results may provide the first evidence that EV71 induces the formation of autolysosomes, because red dots in the 12 h EV71 infected group, which represented autolysosomes, increased significantly compared with the mock group and 8 h infected group in Figure 1A. The detailed mechanism of how EV71 activates autophagy and makes use of autophagy for its own benefit is not well understood, and the relationship between other enteroviruses, such as poliovirus and CVB3, and autophagy is also different than EV71’s relationship with autophagy. In CVB3 infected HEK293 and Hela cells, the formation of a double-membranous autophagosome-like vesicle was induced, but SQSTM1 was accumulated and the fusion of autophagosome with lysosome or degradation in autolysosome was inhibited, suggesting autophagic flux is blocked in CVB3 infected cells. Likewise, inhibition of the autophagy pathway by antagonists, e.g., 3-MA, reduced viral production of CVB3. Conversely, nutrient deprivation (HBSS incubation) or mTOR inhibitor (rapamycin) enhanced CVB3 propagation [76,77,78,79,80,81]. Similar to EV71 and CVB3, poliovirus infection was found to recruit LC3 to the viral replication complex, which was also co-localized with LAMP1 (autolysosome/lysosome marker). Likewise, knockdown of autophagy-related proteins (LC3 and ATG12), or treatment with 3-MA decreased poliovirus viral load, while activation of autophagy by tamoxifen or rapamycin promoted viral production [82]. Recently, two groups have reported completely different roles of ULK1 and FIP200 in poliovirus replication, Corona Velazquez et al. found that it failed to affect poliovirus production, while Abernathy et al. considered that with depletion of ULK1, FIP200 suppresses poliovirus production [73,83]. 

Taken together, though the detailed mechanism of how enterovirus utilizes the autophagy network reported from different research teams is not be perfectly consistent, these works strongly show that enteroviruses are good at manipulating the autophagy network to achieve efficient propagation. Our results further enrich our understanding of the relationship between EV71 and autophagy and possibly provide a new therapeutic target against EV71 infection.

## Figures and Tables

**Figure 1 viruses-12-00756-f001:**
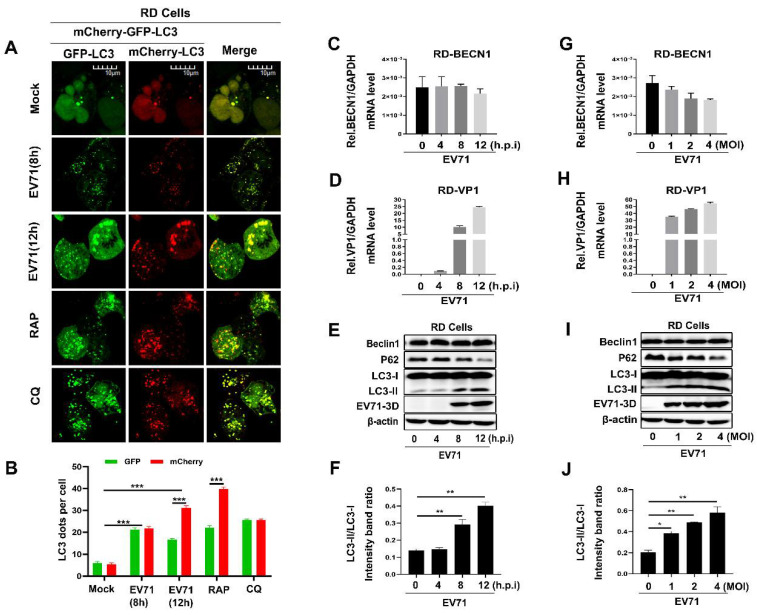
The expression level of Beclin1 was stably maintained during Enterovirus 71 (EV71) infection in rhabdomyosarcoma (RD) cells. (**A**,**B**) RD cells were transfected with GFP-mCherry-LC3 for 24 h, and then infected with EV71 for different times, or treated with Rapamycin (RAP) and Chloroquine diphosphate (CQ) as the complete and incomplete autophagy flux control, respectively. The images were visualized under Laser scanning confocal microscope. Bar = 10 μm (**A**), The graph shows the quantification of autophagosomes by taking the average number of dots in 50 cells (**B**). (**C**–**F**) RD cells were infected with EV71 (MOI = 1) for indicated time, the relative RNA levels of BECN1 (**C**) and EV71-VP1 (**D**) were determined by Real-time PCR, and the relative protein levels of Beclin1 and EV71 3D were detected by Western blot (**E**), and the LC3-II/LC3-I ratio was measured (**F**). (**G**–**J**) RD cells were infected with EV71 for 12 h at an indicated multiplicity of infection (MOI) (0, 1, 2, and 4), the relative RNA levels of BECN1 (**G**) and EV71 VP1 (**H**) were determined by Real-time PCR, the relative protein levels of Beclin1 and EV71-3D were detected by Western blot (**I**), and the LC3-II/LC3-I ratio was measured (**J**). Graph expressed as mean ± SD, ns, not-significant; * *p* < 0.05; ** *p* < 0.01; *** *p* < 0.001.

**Figure 2 viruses-12-00756-f002:**
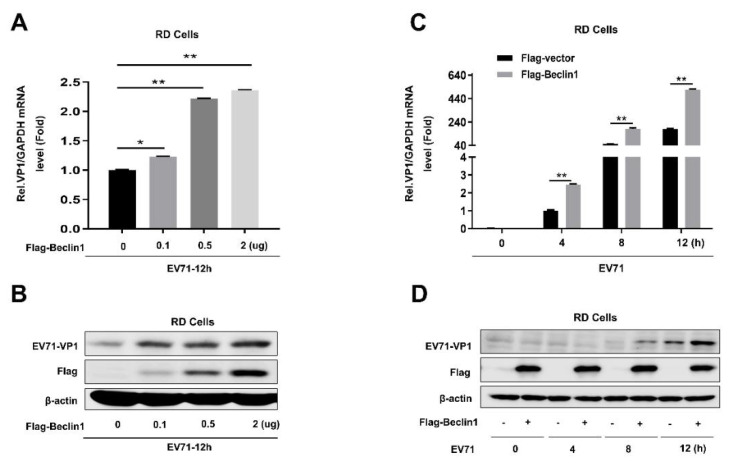
Beclin1 promotes EV71 replication in RD cells. (**A** and **B**) RD cells were transfected with Flag-vector or Flag-Beclin1 at the indicated concentrations for 24 h and infected with EV71 (MOI = 1) for 12 h. The RNA and protein levels of EV71 VP1 were detected by Real-time PCR with *GAPDH* as an internal control (**A**) and Western blot (**B**), respectively. (**C** and **D**) RD cells were transfected with Flag-vector or Flag-Beclin1 for 24 h and then infected with EV71 (MOI = 1) at the indicated times. The RNA and protein levels of EV71-VP1 were detected by Real-time PCR with *GAPDH* as an internal control (**C**) and Western blot (**D**), respectively. Graph expressed as mean ± SD, ns, not-significant; * *p* < 0.05; ** *p* < 0.01.

**Figure 3 viruses-12-00756-f003:**
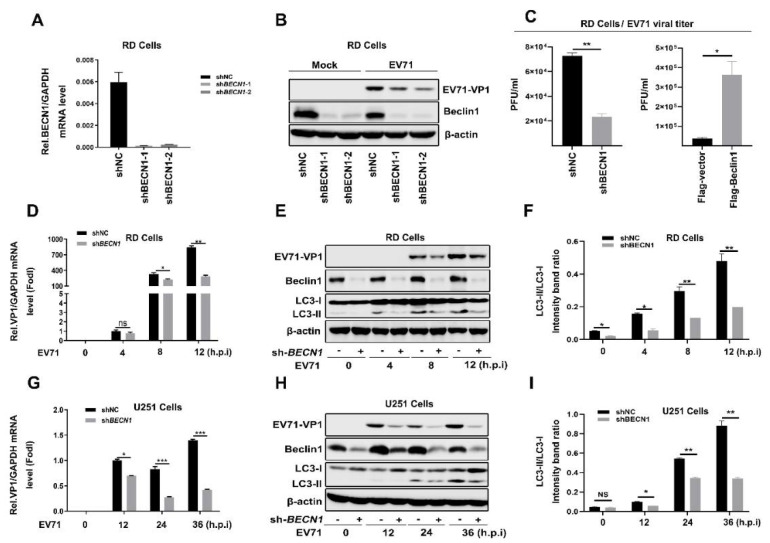
Knockdown of BECN1 attenuates EV71 replication in RD and U251 cells. (**A**,**B**) RD cells were transfected with pLKO.1-shNC, pLKO.1-shBeclin1-1, pLKO.1-shBeclin1-2, and the efficiency of knockdown was testified by Real-time PCR (**A**). The transfected RD cells were infected with EV71 (MOI = 1), and the protein levels of EV71-VP1 and Beclin1 were detected by Western blot with β-actin as an internal control (**B**). (**C**) RD cells were transfected with Flag-vector or Flag-Beclin1, pLKO.1-shNC or pLKO.1-shBeclin1 and infected with EV71 for 12h, and supernatants were collected for TCID50 assay. (**D**–**F**) The stable RD cells were infected with EV71 (MOI = 1) for different periods. The RNA and protein levels of EV71-VP1 were detected by Real-time PCR with *GAPDH* (**D**) and Western blot with β-actin as controls (**E**), and the LC3-II/LC3-I ratio was measured(**F**), respectively. (**G**–**I**) The stable U251 cells were infected with EV71 (MOI = 2) for different periods. The RNA and protein levels of EV71 VP1 were detected by Real-time PCR (**G**) and Western blot (**H**), and the LC3-II/LC3-I ratio was measured(**I**), respectively. Graph expressed as mean ± SD, ns, not-significant; * *p* < 0.05; ** *p* < 0.01; *** *p* < 0.001.

**Figure 4 viruses-12-00756-f004:**
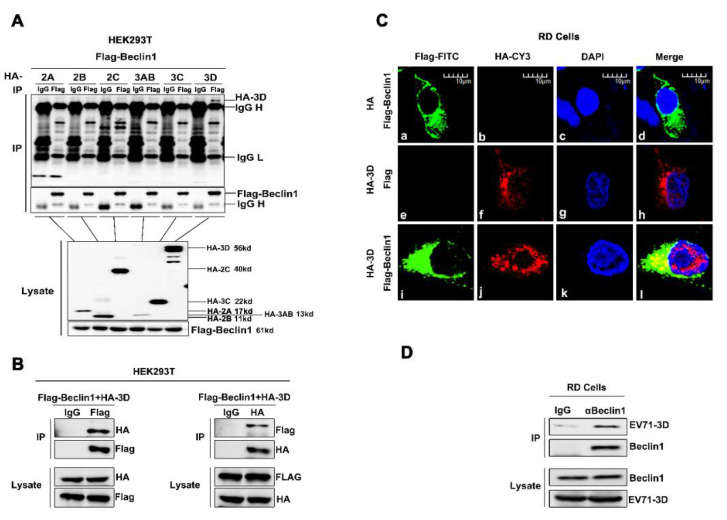
EV71 3D interacts and co-localizes with Beclin1 in the cytoplasm. (**A**) HEK293T cells were co-transfected with plasmid encoding Flag-Beclin1, and plasmids encoding EV71 non-structure proteins, HA-2A, HA-2B, HA-2C, HA-3AB, HA-3C and HA-3D, for 36 h, respectively. The cell lysates were immunoprecipitated with control IgG or an anti-Flag antibody, and then immunoblotted with the indicated antibodies. (**B**) HEK293T cells were co-transfected with plasmids encoding HA-3D and Flag-Beclin1. The cell lysates were immunoprecipitated with control IgG or an anti-Flag antibody or anti-HA antibody, and then immunoblotted with the indicated antibodies. (**C**) RD cells were transfected with plasmids encoding Flag-Beclin1 alone or HA-3D alone or co-transfected with Flag-Beclin1 and HA-3D for 24 h. The localization and distribution of Flag-Beclin1 protein (green), EV71 3D protein (red), and the nuclei (blue) were visualized under Laser scanning confocal microscope. Bar = 10 μm. (**D**) RD cells were infected with EV71 (MOI = 2) for 12 h, and the cells were lysed and immunoprecipitated with control IgG or an anti-Beclin1 antibody and then immunoblotted with indicated antibodies.

**Figure 5 viruses-12-00756-f005:**
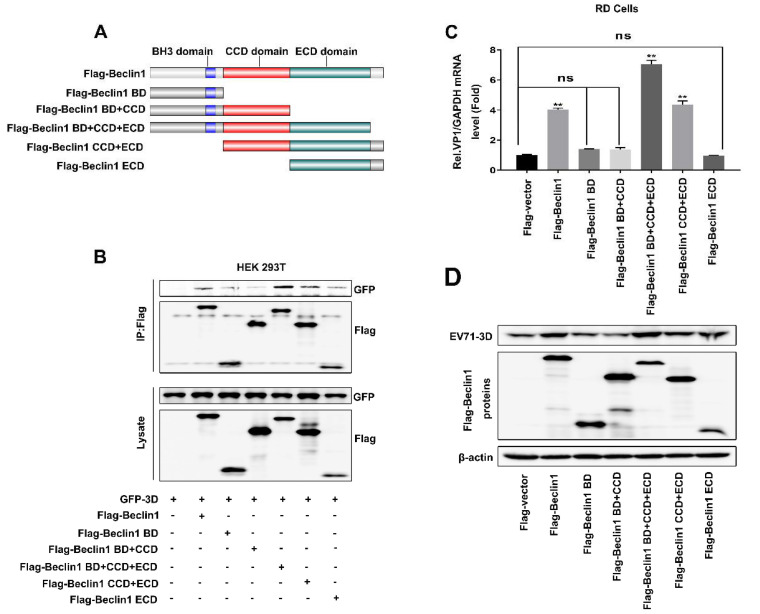
The coiled-coiled domain (CCD) and evolutionary conserved domain (ECD) of Beclin1 were essential for the facilitation of EV71 replication. (**A**) Schematic diagram of Beclin1 and its mutants. (**B**) HEK293T cells were co-transfected with plasmids encoding GFP-3D and Flag-vector as a negative control: Flag-Beclin1, Flag-Beclin1 BD, Flag-Beclin1 BD+CCD, Flag-Beclin1 BD+CCD+ECD, Flag-Beclin1 CCD+ECD, Flag-Beclin1 ECD. The cell lysates were immunoprecipitated with an anti-Flag antibody and then immunoblotted with the indicated antibodies. (**C** and **D**) RD cells were transfected with plasmids expressing Flag-vector, Flag-Beclin1 and various mutants for 24 h and infected with EV71 (MOI = 2) for another 12 h. The expression RNA level of EV71-*VP1* was detected by Real-time PCR with GAPDH as a control (**C**). The protein level of EV71-3D was detected by Western blot (**D**). Graph expressed as mean ± SD, ns, not-significant, ** *p* < 0.01.

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
