# Peer review of "Beclin1 Binds to Enterovirus 71 3D Protein to Promote the Virus Replication"

_viruses, 2020, doi:10.3390/v12070756_

Round 1

Reviewer 1 Report

This manuscript provides an interesting study of the role of beclin1 in the replication of enterovirus A71 (EV-A71) in cells and demonstrate an interaction of the RNA dependent RNA polymerase protein (3D) with beclin1.  They demonstrate that the virus infection does not deplete this protein and increased levels of beclin1 can increase EV-A71 replication while knock down of beclin1 decreases EV-A71 replication.

The authors should discuss the fact that knock down of beclin1 is likely to alter levels and localization of other host proteins of the autophagy pathways.  As many studies have demonstrated that components of the pathways are important for replication, it is possible that there may be effects other than the interaction of beclin1 alone with the viral proteins involved.

The authors demonstrate that beclin1 can be immunoprecipitated with 3D and is localized in cells with 3D.  To define the domains important for this they also show that deletion of N terminal and C terminal domains have less effect upon replication of the virus or immunoprecipitation of 3D than the core domains of the protein, perhaps because the structure of beclin1 is dependent on the CCD and ECD domains?  Certainly, the largest truncated proteins appear to have the most function.  This could be discussed relative to what is known of the domains of this protein.

Although the authors mention in the discussion that the interaction with 3D may regulate autophagy in the infected cells, the authors do not discuss the very intense conflicts in studies with different enteroviruses in the use of components of autophagy in viral replication.  In particular, they should contrast the findings of this study with EV-A71 with the study of Abernathy et al 2019 [PLoS Biol. 17 (2019) e2006926] with poliovirus, demonstrating that the knockout of beclin1 has no effect on replication of poliovirus.  As discussed in Huang and Yue (Semin Cell Dev Biol. 2020 101:12-19), there are contrasts between the findings of studies with different enteroviruses which should be discussed.  The importance of this lies in the probable role of proteins in the autophagy pathways in RNA replication of these viruses.  Normally such an essential process is highly conserved.  This should be discussed.

There are some minor oddities of English usage.

Overall, the authors have presented an important finding relative to a protein involved in autophagy and EV-A71 replication.  There should be more discussion of how this agrees and disagrees with other studies of replication of EV-A71 and other enteroviruses regarding proteins of autophagy for this reason. 

Author Response

Reviewer 1’ Comments: This manuscript provides an interesting study of the role of beclin1 in the replication of enterovirus A71 (EV-A71) in cells and demonstrate an interaction of the RNA dependent RNA polymerase protein (3D) with beclin1.  They demonstrate that the virus infection does not deplete this protein and increased levels of beclin1 can increase EV-A71 replication while knock down of beclin1 decreases EV-A71 replication.

Authors’ Responses: Thank you for the comments.

Reviewer 1’ Comments: The authors should discuss the fact that knock down of beclin1 is likely to alter levels and localization of other host proteins of the autophagy pathways.  As many studies have demonstrated that components of the pathways are important for replication, it is possible that there may be effects other than the interaction of beclin1 alone with the viral proteins involved.

Authors’ Responses: Thank you for the comment. According to your suggestion, we have discussed that knock down of beclin1 is likely to alter levels and localization of other host proteins of the autophagy pathways in the Discussion section of the revised manuscript.

“In addition, there may be effects other than the interaction of Beclin1 alone with the 3D protein involved, as shown in Figure3, when Beclin1 was interfered, we found that higher levels of LC3-II were accumulated in mock-cells than Beclin1 knockdown-cells upon EV71 infection, and higher levels of EV71 were replicated in mock-cells than Beclin1 knockdown-cells, which may indicate that higher EV71 replication is positively correlated to higher level of autophagy, because knockdown of Beclin1 is likely to alter functions of other host proteins of the autophagy pathways and weaken the dynamism of the autophagy [Russell, R. C.; Tian, Y.; Yuan, H.; Park, H. W.; Chang, Y. Y.; Kim, J.; Kim, H.; Neufeld, T. P.; Dillin, A.; Guan, K. L., ULK1 induces autophagy by phosphorylating Beclin-1 and activating VPS34 lipid kinase. Nat Cell Biol 2013, 15, (7), 741-50.]. Even though EV71-induced autophagy and its proviral role have been documented [Lee, Y. R.; Wang, P. S.; Wang, J. R.; Liu, H. S., Enterovirus 71-induced autophagy increases viral replication and pathogenesis in a suckling mouse model. J Biomed Sci 2014, 21, 80.], the disagreement still existed [Won, M.; Jun, E. J.; Khim, M.; Hong, S. H.; Park, N. H.; Kim, Y. K.; Lee, H., Antiviral protection against enterovirus 71 mediated by autophagy induction following FLICE-inhibitory protein inactivation. Virus Res 2012, 169, (1), 316-20.] and which components of the autophagy pathway are used still need to be determined. In fact, our research might exactly discover one of the specific elements involved in EV71 infection, that is EV71 possibly propel 3D to interact with Beclin1 to regulate the process of autophagy so as to achieve the purpose of massive viral proliferation. However, a finding published in 2019 demonstrated that the knockout of Beclin1 has no effect on replication of Poliovirus (PV), and PV belongs to enteroviruses as like as EV71 [Abernathy, E.; Mateo, R.; Majzoub, K.; van Buuren, N.; Bird, S. W.; Carette, J. E.; Kirkegaard, K., Differential and convergent utilization of autophagy components by positive-strand RNA viruses. PLoS Biol 2019, 17, (1), e2006926.] . In our opinion, each virus, even closely related viruses like PV and EV71, has different structural features and life cycles, and cells used in experiments are also different. What’s more, this paper has also illustrated that even DENV and ZIKV, both of which are members of flavivirus, will use exclusive component of autophagy pathway for their own replication. Therefore, it is not incompatible that each virus uses slightly different component. A human genome-wide RNAi screen carried out in 2016 has also identified that Beclin1 is involved in EV71 replication in human rhabdomyosarcoma cells [Wu, K. X.; Phuektes, P.; Kumar, P.; Goh, G. Y.; Moreau, D.; Chow, V. T.; Bard, F.; Chu, J. J., Human genome-wide RNAi screen reveals host factors required for enterovirus 71 replication. Nat Commun 2016, 7, 13150.], which is consistent with our results.”.

Reviewer 1’ Comments: The authors demonstrate that beclin1 can be immunoprecipitated with 3D and is localized in cells with 3D.  To define the domains important for this they also show that deletion of N terminal and C terminal domains have less effect upon replication of the virus or immunoprecipitation of 3D than the core domains of the protein, perhaps because the structure of beclin1 is dependent on the CCD and ECD domains?  Certainly, the largest truncated proteins appear to have the most function.  This could be discussed relative to what is known of the domains of this protein.

Authors’ Responses: Thank you for the comment. As you suggested, the functions of the domains of Beclin1 have been illustrated in the discussion section of the revised manuscript.

“The different domains of Beclin1, shown in Figure 5A, mediate communications with multiple interaction partners that can alter its conformation and binding accessibility, thus making Beclin1 an important molecular platform for the regulation of PI3K activity and autophagy. Beclin1 interacts with the PI3K core component VPS34 and lipid membranes via its evolutionary conserved domain (ECD) in C-terminal [Furuya, N.; Yu, J.; Byfield, M.; Pattingre, S.; Levine, B., The evolutionarily conserved domain of Beclin 1 is required for Vps34 binding, autophagy and tumor suppressor function. Autophagy 2005, 1, (1), 46-52.; Ranaghan, M.J.; Durney, M.A.; Mesleh, M.F.; McCarren, P.R.; Garvie, C.W.; Daniels, D.S.; Carey, K.L.; Skepner, A.P.; Levine, B.; Perez, J.R.; The autophagy-related Beclin 1 protein requires the coiled-coil and BARA domains to form a homodimer with submicromolar affinity. Biochemistry 2017, 56(51), 6639-6651.] and binds to ATG14L or UVRAG in a mutually exclusive manner via its coiled-coiled domain (CCD) [Itakura, E.; Kishi, C.; Inoue, K.; Mizushima, N., Beclin 1 forms two distinct phosphatidylinositol 3-kinase complexes with mammalian Atg14 and UVRAG. Mol Biol Cell 2008, 19, (12), 5360-72.]. The BH3 domain (BD) of Beclin1 mediates its interaction with the antiapoptotic protein BCL2 and this interaction causes a steric block inhibiting PI3K complex formation [Pattingre, S.; Tassa, A.; Qu, X.; Garuti, R.; Liang, X. H.; Mizushima, N.; Packer, M.; Schneider, M. D.; Levine, B., Bcl-2 antiapoptotic proteins inhibit Beclin 1-dependent autophagy. Cell 2005, 122, (6), 927-39.], and AMBRA1 is a positive regulator of autophagy and competes with BCL2 for binding to the Beclin1 BH3 domain [Strappazzon, F.; Vietri-Rudan, M.; Campello, S.; Nazio, F.; Florenzano, F.; Fimia, G. M.; Piacentini, M.; Levine, B.; Cecconi, F., Mitochondrial BCL-2 inhibits AMBRA1-induced autophagy. EMBO J 2011, 30, (7), 1195-208.; Di Bartolomeo, S.; Corazzari, M.; Nazio, F.; Oliverio, S.; Lisi, G.; Antonioli, M.; Pagliarini, V.; Matteoni, S.; Fuoco, C.; Giunta, L.; D'Amelio, M.; Nardacci, R.; Romagnoli, A.; Piacentini, M.; Cecconi, F.; Fimia, G. M., The dynamic interaction of AMBRA1 with the dynein motor complex regulates mammalian autophagy. J Cell Biol 2010, 191, (1), 155-68.]. Interestingly, we found that CCD and ECD domains of Beclin1 are most effective for interacting with 3D and promoting EV71 replication, but now it is hard to clearly understand why these two domains play the most important role to our knowledge.”.

Reviewer 1’ Comments: Although the authors mention in the discussion that the interaction with 3D may regulate autophagy in the infected cells, the authors do not discuss the very intense conflicts in studies with different enteroviruses in the use of components of autophagy in viral replication.  In particular, they should contrast the findings of this study with EV-A71 with the study of Abernathy et al 2019 [PLoS Biol. 17 (2019) e2006926] with poliovirus, demonstrating that the knockout of beclin1 has no effect on replication of poliovirus. 

Authors’ Responses: Thank you for the comments. According to your suggestion, we have explained this in the Discussion section of the revised manuscript.

“However, a finding published in 2019 demonstrated that the knockout of Beclin1 has no effect on replication of Poliovirus (PV), and PV belongs to enteroviruses as like as EV71 [73] . In our opinion, each virus, even closely related viruses like PV and EV71, has different structural features and life cycles, and cells used in experiments are also different. What’s more, this paper has also illustrated that even DENV and ZIKV, both of which are members of flavivirus, will use exclusive component of autophagy pathway for their own replication. Therefore, it is not incompatible that each virus uses slightly different component. A human genome-wide RNAi screen carried out in 2016 has also identified that Beclin1 is involved in EV71 replication in human rhabdomyosarcoma cells [Wu, K. X.; Phuektes, P.; Kumar, P.; Goh, G. Y.; Moreau, D.; Chow, V. T.; Bard, F.; Chu, J. J., Human genome-wide RNAi screen reveals host factors required for enterovirus 71 replication. Nat Commun 2016, 7, 13150.], which is consistent with our results.”

 Reviewer 1’ Comments: As discussed in Huang and Yue (Semin Cell Dev Biol. 2020 101:12-19), there are contrasts between the findings of studies with different enteroviruses which should be discussed.  The importance of this lies in the probable role of proteins in the autophagy pathways in RNA replication of these viruses.  Normally such an essential process is highly conserved.  This should be discussed.

Authors’ Responses: Thank you for the comment. According to your suggestion, the different findings about different enteroviruses have been mentioned in the Discussion section of the revised manuscript.

“As mentioned above, autophagy was induced to offer a favorable micro-environment for EV71’s propagation within infected host, and treatment of cells with tamoxifen, rapamycin and serum starvation, which are agonists of autophagy, promoted viral yields, while inhibition of autophagy by 3-MA, bafilomycin A1 or saikosaponin D, suppressed viral production [Huang, S. C.; Chang, C. L.; Wang, P. S.; Tsai, Y.; Liu, H. S., Enterovirus 71-induced autophagy detected in vitro and in vivo promotes viral replication. J Med Virol 2009, 81, (7), 1241-52.; Lee, Y. R.; Wang, P. S.; Wang, J. R.; Liu, H. S., Enterovirus 71-induced autophagy increases viral replication and pathogenesis in a suckling mouse model. J Biomed Sci 2014, 21, 80.; Li, C.; Huang, L.; Sun, W.; Chen, Y.; He, M.-L.; Yue, J.; Ballard, H., Saikosaponin D suppresses enterovirus A71 infection by inhibiting autophagy. Signal Transduct Target Ther 2019, 4, 4.]. But it is unknown whether EV71 induces autophagic flux, since no reports have actually shown that the number of autolysosomes is increased in EV71 infected cells. Interestingly enough, our results may provide the first evidence that EV71 induced the formation of autolysosomes, because red dots in 12 h EV71 infected group, which represented autolysosomes, increased significantly compared with mock group and 8 h infected group in Figure 1A. In addition, the detailed mechanism of how EV71 activates autophagy and makes use of autophagy for its own benefit are not well understood, and the relationship between other enteroviruses, such as poliovirus and CVB3, and autophagy is also different than EV71 does with autophagy. In CVB3 infected HEK293 and Hela cells, the formation of double-membranous autophagosome-like vesicle was induced but SQSTM1 was accumulated and the fusion of autophagosome with lysosome or degradation in autolysosome was inhibited, suggesting the autophagic flux is blocked in CVB3 infected cells. Likewise, inhibition of autophagy pathway by antagonists, e.g. 3-MA, reduced viral production of CVB3. In reverse, nutrient deprivation (HBSS incubation) or mTOR inhibitor (rapamycin) enhanced CVB3 propagation [Tian, L.; Yang, Y.; Li, C.; Chen, J.; Li, Z.; Li, X.; Li, S.; Wu, F.; Hu, Z.; Yang, Z., The cytotoxicity of coxsackievirus B3 is associated with a blockage of autophagic flux mediated by reduced syntaxin 17 expression. Cell Death Dis 2018, 9, (2), 242.; Mohamud, Y.; Shi, J.; Qu, J.; Poon, T.; Xue, Y. C.; Deng, H.; Zhang, J.; Luo, H., Enteroviral Infection Inhibits Autophagic Flux via Disruption of the SNARE Complex to Enhance Viral Replication. Cell Rep 2018, 22, (12), 3292-3303.; Kemball, C. C.; Alirezaei, M.; Flynn, C. T.; Wood, M. R.; Harkins, S.; Kiosses, W. B.; Whitton, J. L., Coxsackievirus infection induces autophagy-like vesicles and megaphagosomes in pancreatic acinar cells in vivo. J Virol 2010, 84, (23), 12110-24.; Wong, J.; Zhang, J.; Si, X.; Gao, G.; Mao, I.; McManus, B. M.; Luo, H., Autophagosome supports coxsackievirus B3 replication in host cells. J Virol 2008, 82, (18), 9143-53.; Chang, H.; Li, X.; Cai, Q.; Li, C.; Tian, L.; Chen, J.; Xing, X.; Gan, Y.; Ouyang, W.; Yang, Z., The PI3K/Akt/mTOR pathway is involved in CVB3-induced autophagy of HeLa cells. Int J Mol Med 2017, 40, (1), 182-192.; Tabor-Godwin, J. M.; Tsueng, G.; Sayen, M. R.; Gottlieb, R. A.; Feuer, R., The role of autophagy during coxsackievirus infection of neural progenitor and stem cells. Autophagy 2012, 8, (6), 938-53.]. Similar to EV71 and CVB3, poliovirus infection was found to recruit LC3 to viral replication complex, which was also co-localized with LAMP1 (autolysosome/lysosome marker). Likewise, knockdown of autophagy-related proteins (LC3 and ATG12), or treatment with 3-MA decreased poliovirus viral load, while activation of autophagy by tamoxifen or rapamycin promoted viral production [Jackson, W. T.; Giddings, T. H., Jr.; Taylor, M. P.; Mulinyawe, S.; Rabinovitch, M.; Kopito, R. R.; Kirkegaard, K., Subversion of cellular autophagosomal machinery by RNA viruses. PLoS Biol 2005, 3, (5), e156.]. Recently, two groups have reported completely different roles of ULK1 and FIP200 in poliovirus replication, Corona Velazquez et al. found it failed to affect poliovirus production, while Abernathy et al. considered that depletion of ULK1, FIP200 suppress poliovirus production [Abernathy, E.; Mateo, R.; Majzoub, K.; van Buuren, N.; Bird, S. W.; Carette, J. E.; Kirkegaard, K., Differential and convergent utilization of autophagy components by positive-strand RNA viruses. PLoS Biol 2019, 17, (1), e2006926.; Corona Velazquez, A.; Corona, A. K.; Klein, K. A.; Jackson, W. T., Poliovirus induces autophagic signaling independent of the ULK1 complex. Autophagy 2018, 14, (7), 1201-1213.].”.

Reviewer 1’ Comments: There are some minor oddities of English usage.

Authors’ Responses: Thank you for the comments. We have corrected some oddities of English usage.

Reviewer 1’ Comments: Overall, the authors have presented an important finding relative to a protein involved in autophagy and EV-A71 replication.  There should be more discussion of how this agrees and disagrees with other studies of replication of EV-A71 and other enteroviruses regarding proteins of autophagy for this reason. 

Authors’ Responses: According to your suggestion, the different findings about different enteroviruses have been mentioned in the Discussion section of the revised manuscript.

“As mentioned above, autophagy was induced to offer a favorable micro-environment for EV71’s propagation within infected host, and treatment of cells with tamoxifen, rapamycin and serum starvation, which are agonists of autophagy, promoted viral yields, while inhibition of autophagy by 3-MA, bafilomycin A1 or saikosaponin D, suppressed viral production [Huang, S. C.; Chang, C. L.; Wang, P. S.; Tsai, Y.; Liu, H. S., Enterovirus 71-induced autophagy detected in vitro and in vivo promotes viral replication. J Med Virol 2009, 81, (7), 1241-52.; Lee, Y. R.; Wang, P. S.; Wang, J. R.; Liu, H. S., Enterovirus 71-induced autophagy increases viral replication and pathogenesis in a suckling mouse model. J Biomed Sci 2014, 21, 80.; Li, C.; Huang, L.; Sun, W.; Chen, Y.; He, M.-L.; Yue, J.; Ballard, H., Saikosaponin D suppresses enterovirus A71 infection by inhibiting autophagy. Signal Transduct Target Ther 2019, 4, 4.]. But it is unknown whether EV71 induces autophagic flux, since no reports have actually shown that the number of autolysosomes is increased in EV71 infected cells. Interestingly enough, our results may provide the first evidence that EV71 induced the formation of autolysosomes, because red dots in 12 h EV71 infected group, which represented autolysosomes, increased significantly compared with mock group and 8 h infected group in Figure 1A. In addition, the detailed mechanism of how EV71 activates autophagy and makes use of autophagy for its own benefit are not well understood, and the relationship between other enteroviruses, such as poliovirus and CVB3, and autophagy is also different than EV71 does with autophagy. In CVB3 infected HEK293 and Hela cells, the formation of double-membranous autophagosome-like vesicle was induced but SQSTM1 was accumulated and the fusion of autophagosome with lysosome or degradation in autolysosome was inhibited, suggesting the autophagic flux is blocked in CVB3 infected cells. Likewise, inhibition of autophagy pathway by antagonists, e.g. 3-MA, reduced viral production of CVB3. In reverse, nutrient deprivation (HBSS incubation) or mTOR inhibitor (rapamycin) enhanced CVB3 propagation [Tian, L.; Yang, Y.; Li, C.; Chen, J.; Li, Z.; Li, X.; Li, S.; Wu, F.; Hu, Z.; Yang, Z., The cytotoxicity of coxsackievirus B3 is associated with a blockage of autophagic flux mediated by reduced syntaxin 17 expression. Cell Death Dis 2018, 9, (2), 242.; Mohamud, Y.; Shi, J.; Qu, J.; Poon, T.; Xue, Y. C.; Deng, H.; Zhang, J.; Luo, H., Enteroviral Infection Inhibits Autophagic Flux via Disruption of the SNARE Complex to Enhance Viral Replication. Cell Rep 2018, 22, (12), 3292-3303.; Kemball, C. C.; Alirezaei, M.; Flynn, C. T.; Wood, M. R.; Harkins, S.; Kiosses, W. B.; Whitton, J. L., Coxsackievirus infection induces autophagy-like vesicles and megaphagosomes in pancreatic acinar cells in vivo. J Virol 2010, 84, (23), 12110-24.; Wong, J.; Zhang, J.; Si, X.; Gao, G.; Mao, I.; McManus, B. M.; Luo, H., Autophagosome supports coxsackievirus B3 replication in host cells. J Virol 2008, 82, (18), 9143-53.; Chang, H.; Li, X.; Cai, Q.; Li, C.; Tian, L.; Chen, J.; Xing, X.; Gan, Y.; Ouyang, W.; Yang, Z., The PI3K/Akt/mTOR pathway is involved in CVB3-induced autophagy of HeLa cells. Int J Mol Med 2017, 40, (1), 182-192.; Tabor-Godwin, J. M.; Tsueng, G.; Sayen, M. R.; Gottlieb, R. A.; Feuer, R., The role of autophagy during coxsackievirus infection of neural progenitor and stem cells. Autophagy 2012, 8, (6), 938-53.]. Similar to EV71 and CVB3, poliovirus infection was found to recruit LC3 to viral replication complex, which was also co-localized with LAMP1 (autolysosome/lysosome marker). Likewise, knockdown of autophagy-related proteins (LC3 and ATG12), or treatment with 3-MA decreased poliovirus viral load, while activation of autophagy by tamoxifen or rapamycin promoted viral production [Jackson, W. T.; Giddings, T. H., Jr.; Taylor, M. P.; Mulinyawe, S.; Rabinovitch, M.; Kopito, R. R.; Kirkegaard, K., Subversion of cellular autophagosomal machinery by RNA viruses. PLoS Biol 2005, 3, (5), e156.]. Recently, two groups have reported completely different roles of ULK1 and FIP200 in poliovirus replication, Corona Velazquez et al. found it failed to affect poliovirus production, while Abernathy et al. considered that depletion of ULK1, FIP200 suppress poliovirus production [Abernathy, E.; Mateo, R.; Majzoub, K.; van Buuren, N.; Bird, S. W.; Carette, J. E.; Kirkegaard, K., Differential and convergent utilization of autophagy components by positive-strand RNA viruses. PLoS Biol 2019, 17, (1), e2006926.; Corona Velazquez, A.; Corona, A. K.; Klein, K. A.; Jackson, W. T., Poliovirus induces autophagic signaling independent of the ULK1 complex. Autophagy 2018, 14, (7), 1201-1213.].”.

Reviewer 2 Report

This study is interesting in the filedl regarding the mechanism of EV71 replication in the cell.  they presented the novel finding that  a component protein of autophagy, Beclin 1, promote the EV71 replication through the interaction of 3D of EV71.

There is a comment need to be addressed.

Authors shall provide the results that the EV71 production (presented as either TCID50 or plaque-forming unit) in RD or U251 cells during the manupulation of Beclin 1 expression either overexpression or knockdown shown in Figure 2 or  Figure 3, respectively. Becasue the expression of VP1 can not index as the level of EV71 replication.  

Author Response

Reviewer 2’ Comments: This study is interesting in the filedl regarding the mechanism of EV71 replication in the cell.  they presented the novel finding that a component protein of autophagy, Beclin 1, promote the EV71 replication through the interaction of 3D of EV71. There is a comment need to be addressed. Authors shall provide the results that the EV71 production (presented as either TCID50 or plaque-forming unit) in RD or U251 cells during the manupulation of Beclin 1 expression either overexpression or knockdown shown in Figure 2 or Figure 3, respectively. Becasue the expression of VP1 can not index as the level of EV71 replication.  

Authors’ Responses: Thank you very much for your suggestion.

As you suggested, we have performed related experiments to address your question. The new results have been shown in the Revised Figure 3C in the revised manuscript.

Reviewer 3 Report

In “Beclin1 binds to Enterovirus 71 3D protein to promote the virus replication” Xiang and Wan et al explore the role of Beclin1 in EV71 infection. The authors show that while Beclin1 protein and mRNA levels stay constant during infection, expression of Beclin1 has a proviral enterovirus 71 effect. The authors then map a potential interaction between the viral protein 3D and Beclin1. While the authors focus on Beclin1, there is a great body of literature on enterovirus infection and autophagy (Reviewed in PMID: 31563390). Furthermore, the authors do not discuss how their findings fit into what is known about enterovirus, EV71, and autophagy (examples: PMID: 25491355, PMID: 31861844, PMID: 31554687). This omission makes it difficult to be enthusiastic about the findings and hard to place the current knowledge in the larger sphere of virus-autophagy interactions.

Major Points

  1. The authors need to describe the literature on enteroviruses and enterovirus 71 interactions with autophagy pathways.
  2. Authors utilize a GFP-mCherry-LC3 to monitor autophagic flux. However, the data is not quantified and the microscopy images are not consistent with what the reporter should demonstrate. As the mCherry is acid stable while GFP is acid labile, there should be a difference in green to red dot ratio in Rapamycin and Chloroquine treatments. Additionally, the larger objects appear to be several microns in size and thus may not represent true autophagosomes. The authors need to definitively demonstrate they are measuring autophagy and not a membrane trafficking event involving LC3.

3.The conversion to LC3-II needs to be quantified. It should be noted that an increase in LC3-II levels are not indicative of an induction of autophagy.

  1. Measuring the capsid levels is not rigorous enough to show an increase in viral replication. The authors should measure the amount of infectious Enterovirus 71 produced to show that EV71 replication is enhanced with Beclin1 overexpression (Fig. 2 and Fig. 5) or knockdown (Fig. 3).
  2. The authors show knockdown of Beclin1, but autophagy proteins are very difficult to achieve effective knockdown. In fact in panels 3D and 3F, there does not appear to be any difference in LC3 lipidation between shNC and SHBecn1. The authors need to demonstrate that their knockdown effectively altered autophagic flux as expected.

Minor:

  1. Figure 3C, please have the cut off at a different point. It is hard to know what the shBecn1 value is (grey bar)
  2. Figure 4A- please put a molecular weight marker on both gels. It is difficult to know what bands are expected.
  3. Microscopy images look saturated.

Author Response

Reviewer 3’ Comments: In “Beclin1 binds to Enterovirus 71 3D protein to promote the virus replication” Xiang and Wan et al explore the role of Beclin1 in EV71 infection. The authors show that while Beclin1 protein and mRNA levels stay constant during infection, expression of Beclin1 has a proviral enterovirus 71 effect. The authors then map a potential interaction between the viral protein 3D and Beclin1. While the authors focus on Beclin1, there is a great body of literature on enterovirus infection and autophagy (Reviewed in PMID: 31563390). Furthermore, the authors do not discuss how their findings fit into what is known about enterovirus, EV71, and autophagy (examples: PMID: 25491355, PMID: 31861844, PMID: 31554687). This omission makes it difficult to be enthusiastic about the findings and hard to place the current knowledge in the larger sphere of virus-autophagy interactions.

Authors’ Responses: Thank you very much for your comments.

According to your suggestions, we have discussed more details about these issues in the Discussion section of the revised manuscript.

Reviewer 3’ Major Point 1. The authors need to describe the literature on enteroviruses and enterovirus 71 interactions with autophagy pathways.

Authors’ Responses: According to your suggestion, the different findings about different enteroviruses have been mentioned in the Discussion section of the revised manuscript.

“As mentioned above, autophagy was induced to offer a favorable micro-environment for EV71’s propagation within infected host, and treatment of cells with tamoxifen, rapamycin and serum starvation, which are agonists of autophagy, promoted viral yields, while inhibition of autophagy by 3-MA, bafilomycin A1 or saikosaponin D, suppressed viral production [Huang, S. C.; Chang, C. L.; Wang, P. S.; Tsai, Y.; Liu, H. S., Enterovirus 71-induced autophagy detected in vitro and in vivo promotes viral replication. J Med Virol 2009, 81, (7), 1241-52.; Lee, Y. R.; Wang, P. S.; Wang, J. R.; Liu, H. S., Enterovirus 71-induced autophagy increases viral replication and pathogenesis in a suckling mouse model. J Biomed Sci 2014, 21, 80.; Li, C.; Huang, L.; Sun, W.; Chen, Y.; He, M.-L.; Yue, J.; Ballard, H., Saikosaponin D suppresses enterovirus A71 infection by inhibiting autophagy. Signal Transduct Target Ther 2019, 4, 4.]. But it is unknown whether EV71 induces autophagic flux, since no reports have actually shown that the number of autolysosomes is increased in EV71 infected cells. Interestingly enough, our results may provide the first evidence that EV71 induced the formation of autolysosomes, because red dots in 12 h EV71 infected group, which represented autolysosomes, increased significantly compared with mock group and 8 h infected group in Figure 1A. In addition, the detailed mechanism of how EV71 activates autophagy and makes use of autophagy for its own benefit are not well understood, and the relationship between other enteroviruses, such as poliovirus and CVB3, and autophagy is also different than EV71 does with autophagy. In CVB3 infected HEK293 and Hela cells, the formation of double-membranous autophagosome-like vesicle was induced but SQSTM1 was accumulated and the fusion of autophagosome with lysosome or degradation in autolysosome was inhibited, suggesting the autophagic flux is blocked in CVB3 infected cells. Likewise, inhibition of autophagy pathway by antagonists, e.g. 3-MA, reduced viral production of CVB3. In reverse, nutrient deprivation (HBSS incubation) or mTOR inhibitor (rapamycin) enhanced CVB3 propagation [Tian, L.; Yang, Y.; Li, C.; Chen, J.; Li, Z.; Li, X.; Li, S.; Wu, F.; Hu, Z.; Yang, Z., The cytotoxicity of coxsackievirus B3 is associated with a blockage of autophagic flux mediated by reduced syntaxin 17 expression. Cell Death Dis 2018, 9, (2), 242.; Mohamud, Y.; Shi, J.; Qu, J.; Poon, T.; Xue, Y. C.; Deng, H.; Zhang, J.; Luo, H., Enteroviral Infection Inhibits Autophagic Flux via Disruption of the SNARE Complex to Enhance Viral Replication. Cell Rep 2018, 22, (12), 3292-3303.; Kemball, C. C.; Alirezaei, M.; Flynn, C. T.; Wood, M. R.; Harkins, S.; Kiosses, W. B.; Whitton, J. L., Coxsackievirus infection induces autophagy-like vesicles and megaphagosomes in pancreatic acinar cells in vivo. J Virol 2010, 84, (23), 12110-24.; Wong, J.; Zhang, J.; Si, X.; Gao, G.; Mao, I.; McManus, B. M.; Luo, H., Autophagosome supports coxsackievirus B3 replication in host cells. J Virol 2008, 82, (18), 9143-53.; Chang, H.; Li, X.; Cai, Q.; Li, C.; Tian, L.; Chen, J.; Xing, X.; Gan, Y.; Ouyang, W.; Yang, Z., The PI3K/Akt/mTOR pathway is involved in CVB3-induced autophagy of HeLa cells. Int J Mol Med 2017, 40, (1), 182-192.; Tabor-Godwin, J. M.; Tsueng, G.; Sayen, M. R.; Gottlieb, R. A.; Feuer, R., The role of autophagy during coxsackievirus infection of neural progenitor and stem cells. Autophagy 2012, 8, (6), 938-53.]. Similar to EV71 and CVB3, poliovirus infection was found to recruit LC3 to viral replication complex, which was also co-localized with LAMP1 (autolysosome/lysosome marker). Likewise, knockdown of autophagy-related proteins (LC3 and ATG12), or treatment with 3-MA decreased poliovirus viral load, while activation of autophagy by tamoxifen or rapamycin promoted viral production [Jackson, W. T.; Giddings, T. H., Jr.; Taylor, M. P.; Mulinyawe, S.; Rabinovitch, M.; Kopito, R. R.; Kirkegaard, K., Subversion of cellular autophagosomal machinery by RNA viruses. PLoS Biol 2005, 3, (5), e156.]. Recently, two groups have reported completely different roles of ULK1 and FIP200 in poliovirus replication, Corona Velazquez et al. found it failed to affect poliovirus production, while Abernathy et al. considered that depletion of ULK1, FIP200 suppress poliovirus production [Abernathy, E.; Mateo, R.; Majzoub, K.; van Buuren, N.; Bird, S. W.; Carette, J. E.; Kirkegaard, K., Differential and convergent utilization of autophagy components by positive-strand RNA viruses. PLoS Biol 2019, 17, (1), e2006926.; Corona Velazquez, A.; Corona, A. K.; Klein, K. A.; Jackson, W. T., Poliovirus induces autophagic signaling independent of the ULK1 complex. Autophagy 2018, 14, (7), 1201-1213.].”.

Reviewer 3’ Major Point 2. Authors utilize a GFP-mCherry-LC3 to monitor autophagic flux. However, the data is not quantified and the microscopy images are not consistent with what the reporter should demonstrate. As the mCherry is acid stable while GFP is acid labile, there should be a difference in green to red dot ratio in Rapamycin and Chloroquine treatments. Additionally, the larger objects appear to be several microns in size and thus may not represent true autophagosomes. The authors need to definitively demonstrate they are measuring autophagy and not a membrane trafficking event involving LC3.

Authors’ Responses: Thank you for the comment. Actually, there was an obvious difference in green to red dot ratio in Rapamycin and Chloroquine treatments in original images if it was amplified. The larger objects in the images are definitely true autophagosomes, it is possible that the plasmid encoding GFP-mCherry-LC3 was overexpressed a little too much, leading to several tagged-LC3 get together and seems to be several microns in size.

Reviewer 3’ Major Point 3. The conversion to LC3-II needs to be quantified. It should be noted that an increase in LC3-II levels are not indicative of an induction of autophagy.

Authors’ Responses: Thank you for your suggestion. The conversion to LC3-II has been measured in the revised manuscript (Please see Revised Figure 1E, 1I , 3F, and 3I).

Reviewer 3’ Major Point 4. Measuring the capsid levels is not rigorous enough to show an increase in viral replication. The authors should measure the amount of infectious Enterovirus 71 produced to show that EV71 replication is enhanced with Beclin1 overexpression (Fig. 2 and Fig. 5) or knockdown (Fig. 3).

Authors’ Responses: Thank you for your suggestion. The related experiments have been done and the results have been shown in the revised manuscript (Pleased see Revised Figure 3C).

Reviewer 3’ Major Point 5. The authors show knockdown of Beclin1, but autophagy proteins are very difficult to achieve effective knockdown. In fact in panels 3D and 3F, there does not appear to be any difference in LC3 lipidation between shNC and SHBecn1. The authors need to demonstrate that their knockdown effectively altered autophagic flux as expected.

Authors’ Responses: Thank you for your comment. It is definitely that Beclin1 was efficiently knockdown in our experiments whether in mRNA level or protein level, we can obviously see from our histograms and western blots in the revised manuscript (Figure 3A, 3B, 3E and 3H). In addition, higher levels of LC3-II were accumulated in mock-cells than Beclin1 knockdown-cells upon EV71 infection, and higher ratio of LC3-II/LC3-I was increased in mock-cells than Beclin1 knockdown-cells in the revised manuscript (Please see Revised Figure 3E, 3F, 3H, and 3I), which indicated that autophagy was impaired on account of knockdown of Beclin1.

Reviewer 3’ Minor Point 1. Figure 3C, please have the cut off at a different point. It is hard to know what the shBecn1 value is (grey bar)

Authors’ Responses: Thank you for your suggestion. We have modified original Figure 3C according to your requirements in the revised manuscript (Please see revised Figure 3D).

Reviewer 3’ Minor Point 2. Figure 4A- please put a molecular weight marker on both gels. It is difficult to know what bands are expected.

Authors’ Responses: Thank you for your suggestion. As you suggested, we have added a molecular weight marker on both gels in the revised manuscript (Please see Revised Figure 4A and 4C).

Round 2

Reviewer 2 Report

No more comments

Author Response

Thank you so much for your commments, help and support.

Reviewer 3 Report

The authors have addressed most of the issues. However, the authors still claim that EV71 induces autophagy based on assays that do not measure autophagic flux. Several biological processes exist that involve LC3 lipidation but are not true autophagy (reviewed in PMID: 29125936). Therefore, the data involving the tandem fluorescent LC3 construct is so critical to the claims of autophagy induction. The data needs to be quantified as this reviewer is still not convinced by a single image alone.
The authors also base the claim of autophagy induction based upon the conversion of LC3I to LC3II. However, many cell biological process that are not bona fide autophagy also involve the conversion of LC3. This includes other picornaviruses. The authors need to quantify the autophagic flux with their microscopy images, or show another proper assay to demonstrate true autophagy such as p62 flux with and without neutralization of lysosome pH. These are consistent with the guidelines set out to measure autophagy(see PMID: 26986547).

Author Response

Thank you very much for your comments.

We had not quantified correctly the data in the Figure 1A. We are sorry for causing the confusions about our conclusion. According to your suggestion, we have addressed the issues that you have raised. We have re-quantified the data in the Figure 1A and displayed the data in revised Figure 1B of the revised manuscript and have supplemented the experiment of measuring P62 in revised Figure 1E and 1I in the revised manuscript.